# Investigation of the Release Mechanism and Mould Resistance of Citral-Loaded Bamboo Strips

**DOI:** 10.3390/polym13193314

**Published:** 2021-09-28

**Authors:** Rui Peng, Jingjing Zhang, Chungui Du, Qi Li, Ailian Hu, Chunlin Liu, Shiqin Chen, Yingying Shan, Wenxiu Yin

**Affiliations:** College of Chemistry and Materials Engineering, Zhejiang A & F University, Hangzhou 311300, China; 18255276196@163.com (R.P.); jingjingzhang312@163.com (J.Z.); LQ950011@163.com (Q.L.); hal15857832323@163.com (A.H.); eustaceweaver7187@gmail.com (C.L.); 18768107239@163.com (S.C.); syy15968566686@163.com (Y.S.); yinwenxiu_110@163.com (W.Y.)

**Keywords:** nanohydrogels, citral, sustained-release, mechanism, bamboo, anti-mould

## Abstract

In the present study, the sustained-release system loading citral was synthesised by using PNIPAm nanohydrogel as a carrier and analysed its drug-release kinetics and mechanism. Four release models, namely zero-order, first-order, Higuchi, and Peppas, were employed to fit the experimental data, and the underlying action mechanism was analysed. The optimised system was applied to treat a bamboo mould, followed by assessment of the mould-proof performance. Our experimental results revealed that the release kinetics equation of the system conformed to the first order; the higher the external temperature, the better the match was. In the release process, PNIPAm demonstrated a good protection and sustained-release effect on citral. Under the pressure of 0.5 MPa, immersion time of 120 min, and the system concentration ratio of 1, the optimal drug-loading parameters were obtained using the slow-release system with the best release parameters. Compared to the other conditions, bamboos treated with pressure impregnation demonstrated a better control effect on bamboo mould, while the control effect on *Penicillium citrinum*, *Trichoderma viride*, *Aspergillus niger*, and mixed mould was 100% after 28 days. Moreover, the structure and colour of bamboo remained unchanged during the entire process of mould control.

## 1. Introduction

Parallel to the excessive carbon dioxide emissions, the emission of greenhouse gases has increased considerably, making climate change a global challenge for mankind. Following the ‘carbon neutral’ goal of the Paris Agreement, the global market demand for low-carbon materials has been increasing [1,2]. As a natural low-carbon material, bamboo offers the advantages of sustainable logging and utilisation, high strength to weight ratio, high strength, and biodegradability [3,4,5]. It is not only commonly used in the construction, furniture, and interior decoration industries but also has significant application prospects in the aerospace domain [6,7,8,9]. Therefore, vigorous production and application of bamboo are crucial for promoting the transformation of social development mode from high carbon to green and low carbon. However, bamboo gets easily infected with mould under certain temperature and humidity conditions owing to its high sugar and protein contents, resulting in serious pollution and the loss of bamboo use value [10,11]. Thus, research on bamboo mould prevention can not only effectively improve the utilisation efficiency of bamboo but also play a positive role in realising the vision of a carbon-free future.

Nanomaterials are one of the most widely studied and applied materials owing to their unique nano-properties [12,13,14]. These materials have also become a research hotspot in the field of bamboo mould prevention to prepare a sustained-release system with nanomaterials as a carrier material. Slow release refers to the slowdown of the release rate of loaded fungicides that extends the associated release time under isolation and protection of the carrier material [15,16]. Notably, the release of mould agents is the key to determining the effect of mould. Based on the different properties of the carrier material, such as low critical solution temperature, permeability, viscosity, degradability, and concentration [17,18,19,20], the release parameters, including the release time, release amount, and release rate [21], can be adjusted to a certain extent, which in turn affects the performance of mould prevention in a slow-release system. Hydrophilic carriers are usually selected because they possess pore space to promote the release of mould inhibitors, especially for low-water-soluble drugs, which easily bind with the hydrophilic groups in bamboo. When the carrier comes into contact with the water medium, various phenomena such as degradation, dissolution, and swelling can occur [22]. Under the action of diffusion, convection, explosion, ion exchange, osmotic pressure, and other mechanisms, different release curves are produced [23]. In addition, the release parameters are related to the properties of fungicides, such as the solubility, particle size, and hydrophilicity [15]. The degradation rate of the carrier, the drug load, and the characteristics of the fungicide synergistically influence the release effect of the fungicide. Therefore, according to the choice of carrier and fungicide, different release mechanisms have been found conducive to the release of fungicide from the carrier material. However, depending on the proportion of participation, each mechanism can represent either a major release mechanism or a negligible one.

Although the sustained-release system with mould inhibitor has been applied and assessed in the field of mould control of bamboo, studies have mainly focused on the preparation process and performance, while the systematic studies on the mechanism of sustained release are rare [10,14]. Polyn-isopropylacrylamide (PNIPAm) nanohydrogels are widely used in drug delivery systems as drug delivery carriers due to their good thermal sensitivity and biocompatibility, and their unique amphiphilic groups have a strong solubilizing effect on hydrophobic drugs. The natural antibacterial agent citral was encapsulated in PNIPAm nanohydrogel, which not only overcomes its own disadvantages of easy oxidation and volatile, but also facilitates the control of release, so that its hydrophilicity, chemical stability, antibacterial activity and utilization efficiency have been greatly improved. In this study, a PNIPAm/citral nanohydrogel (P/Cn) sustained-release system was prepared using PNIPAm nanohydrogel as the carrier material and a natural plant essential oil, citral, as the mould inhibitor. To explore the release mechanism of the sustained-release system, the release parameters of P/Cn were investigated through a release experiment, where the release kinetics equation was fitted. On this basis, the technology behind P/Cn for mould prevention treatment of bamboo was optimised, and the mould prevention performance was discussed. The schematic diagram of the experimental principle in this study is shown in Figure 1. The purpose of this study was to provide a theoretical and practical basis for obtaining the ideal release effect of P/Cn as well as to provide new ideas and methods for realising the efficient and lasting protection of bamboo material.

## 2. Materials and Methods

### 2.1. Materials

P/Cns were synthesised in the laboratory. Details of the synthesis are presented in Appendix A. Briefly, disodium hydrogen phosphate (AR) and sodium dihydrogen phosphate (AR) were both sourced from Sinophenol Chemical Reagent Co., Ltd. (Shanghai, China). Polyethylene glycol monostearate (PEGMS, *n* = 45, AR) was supplied by China Aladdin Reagent Co., Ltd. (Shanghai, China). Anhydrous ethanol (AR) was provided by Tianjin Yongda Chemical Reagent Co., Ltd. (Tianjin, China). Bamboo (moisture content of approximately 10%) was purchased from Zhejiang Yongyu Bamboo Industry Co., Ltd. (Hangzhou, China). and customised into a unified specification of 50 × 20 × 5 mm^3^ strips in the laboratory. All reagents and solvents were used directly in this experiment without further purification.

### 2.2. Preparation of Phosphate-Buffered Saline

The preparation process of phosphate-buffered saline (PBS; 0.1 mol/L, pH = 6.8) was as follows: 35.822 g of disodium hydrogen phosphate and 15.603 g of disodium hydrogen phosphate were weighed and added to 2000 mL deionised water. Finally, 2.0 g of PEGMS was added and stirred until a clarified mixture was obtained, which was then stored at room temperature until further use.

### 2.3. The Standard Curve

First, the maximum absorption wavelength (λ_Max_) of citral in PBS was measured by scanning the buffer at a full wavelength on an ultraviolet–visible (UV–Vis) spectrophotometer. Second, 0.01 g of citral was weighed accurately, and a small amount of PBS solution was added to the mixture. The concentrated storage solution of 100 μg/mL was finally obtained by transferring to achieve a constant volume in a 100-mL volumetric flask. Then, 2–8 μg/mL of standard citral solution was prepared in PBS as the solvent. Finally, the absorbance value of each citral standard solution was measured (measured thrice and then averaged) at λ_Max_, and the absorbance was plotted against the concentration to draw a standard curve.

### 2.4. Release and Detection of Citral

Dialysis is the most common method to determine the extent of drug release in vitro in the nano-drug delivery systems [24,25]. Briefly, 1 g of the P/Cn solution was accurately weighed into a dialysis bag (MWCO: 8000–14,000) and placed in a beaker containing 100 mL of PBS. Then, the beaker was placed in a water bath and stirred at a rate of 100 rpm. Afterwards, 2 mL of the solution was removed from the beaker for detection within a specified time. The same volume of fresh PBS buffer was added to the beaker [26].

The absorbance of citral (at λ_Max_) was determined through UV–Vis spectrophotometry after stirring the solution uniformly and adding fresh PBS solution to 20 mL. Then, the cumulative release L_2_ of citral at different time points was calculated from the standard curve (measured thrice and then averaged), and the release rate of citral was calculated according to the following Formula (1):ER (%) = L_2_/L_1_× 100%,(1)
where ER is the release rate of citral, %; L_1_ is the mass of encapsulated citral (see the Appendix A), mg; L_2_ is the cumulative release of citral at different time points, mg.

### 2.5. Optimisation of Release Parameters

The addition of citral (mg), the reaction time (h), and the reaction temperature (°C) were selected as the investigation factors. The orthogonal test design of L_9_(3^4^) was selected to determine the optimal release parameters of P/Cn under the release temperature of 36 °C.

### 2.6. Sustained-Release Kinetics

On the basis of the release experiment, the release curve of P/Cn was plotted. Four release models, including zero-order, first-order, Higuchi, and Peppas, were employed to fit the release curve [27], and the release mechanism was described in detail. The four dynamics equations applied are as follows:Q = K t,(Ze-ro-order)
Q = 1 − e^(−k t)^,(First-order)
Q = K t^1/2^,(Higu-chi)
Q = K t^n^,(Peppas)
where Q and t represent the release rate and release time, respectively; K represents the release rate constant; and n represents the diffusion index.

### 2.7. Impregnation of Bamboo Strips

First, the bamboo strips with a moisture content of approximately 10% were weighed (m_1_) and fully immersed in a beaker containing P/Cn. The strips were then placed in a pressurised tank for the dipping treatment. Then, the impregnated bamboo strips were weighed (m_2_) and placed in a vacuum drying oven. The bamboo strips were dried to approximately 10% moisture content at 30 °C and then weighed (m_3_). Finally, the samples were stored in a dryer for subsequent experiments.

### 2.8. Optimisation of the Impregnation Process

The value of pressure, impregnation time, and P/Cn concentration ratio (CR = P/Cn concentration of the treated bamboo strips divided by the P/Cn original concentration) were selected as experimental factors and used to investigate the influence of various factors on the drug-loading capacity of the bamboo strips. The wet drug load (R_1_) and dry drug load (R_2_) of the sample were calculated using the Formulas (2) and (3), respectively:(2)$$R_{1}=\frac{(m_{2}-m_{1})\times C\times 10^{6}}{S},$$
(3)$$R_{2}=\frac{(m_{3}-m_{1})\times C\times 10^{6}}{S},$$

In the formula, R_1_ and R_2_ indicate wet and dry drug loadings, respectively, mg/m^2^; m_1_, m_2,_ and m_3_ indicate the mass of bamboo strips before treatment, the wet weight after treatment, and the dry weight after drying, respectively, mg; C is the concentration (mass fraction) of citral in the system, %; and S is the sum of the surface area of the six surfaces of the bamboo strips, m^2^.

### 2.9. Fourier-Transformed Infrared Spectroscopy

The bamboo strips were crushed with a micro-plant pulveriser and screened with a 300-purpose screen. The bamboo powder obtained was mixed with potassium bromide and pressed into thin slices according to the mass of 1:100 and finally scanned with IR Prestige-21 (Shimadzu, Shanghai, China). The resolution of Fourier-transform infrared spectroscopy (FT-IR) was kept at 4 cm^−1^, and the wavelength range was 4000–400 cm^−1^.

### 2.10. Anti-Mould Property

The bamboo strips were treated by P/Cn atmospheric pressure impregnation (OT) and pressure impregnation (PT) and treated by P/Cn pressure impregnation after the completion of the release experiment (RT). According to GB/T 18261-2013 [28], an indoor mould test was performed under the conditions of 28 ± 2 °C temperature and 85% ± 5% relative humidity. The control effects of the samples on *Penicillium citrinum* (PC), *Trichoderma viride* (TV), *Aspergillus niger* (AN), and mixed mould (mixed ratio of PC, TV, and AN was 1:1:1, PTA for short) were investigated, and the untreated bamboo strips (UT) were set as the blank control. During the mould prevention period, the degree of mould infection on the bamboo strips was recorded weekly (see the Appendix A for the evaluation criteria of the infection level), and the bamboo strips were photographed after 28 days of mould prevention to analyse the mould prevention results.

## 3. Results and Discussion

### 3.1. Draw the Standard Curve

The maximum absorption peak of citral was obtained at 238 nm (Figure 2A), while P/Cn showed a large absorption value at the same wavelength, which was similar to the maximum absorption wavelength of 240 nm and was not interfered with other substances. Accordingly, the standard solution of citral-PBS with a concentration of 2–8 μg/mL was prepared, and the absorbance value was measured at the wavelength of 238 nm. Figure 2B depicts the regression equation of the standard curve:$$\begin{array}{cc} A\;=0.08200\;C\;-0.00571, & \left(R^{2}=0.999731\right) \end{array}$$

The fitting correlation coefficient R^2^ of the standard curve of citral in PBS was ultra-close to 1, which indicated a favourable linear relationship between the concentration of citral and absorbance. The use of this standard curve can avoid experimental errors to the maximum extent and help meet the experimental requirements.

### 3.2. Orthogonal Test

To obtain the optimal release parameters of P/Cn (L_2_ and ER), orthogonal tests were designed using the L_9_ (3^4^) orthogonal table (See the Appendix A for the factor level), and the results are shown in Table 1. According to the R-value, the influence of each factor on L_2_ was in the order of A > C > B, with the optimal combination at A_3_B_2_C_1_ (P/Cn8). In other words, under the citral amount of 50 mg, reaction time of 2 h, and reaction temperature of 25 °C, the cumulative release of citral was 33.469 mg, and the release rate was 89.986%. Conversely, the influence of each factor on ER was in the order of A > B > C, while the optimal combination was A_3_B_3_C_2_ (P/Cn9). In other words, under the citral amount of 50 mg, reaction time of 3 h, and reaction temperature of 30 °C, the cumulative release amount of citral was 31.093 mg, and the release rate was 95.828%.

Variance analysis was conducted for L_2_ (Table 2). At the significance level of 1%, the amount of citral added showed a significant impact on the cumulative release of citral. As shown in Table 2, the amount of citral added also made a significant difference to the cumulative release of citral under the significance level of 10%. Hence, the amount of citral showed a significant effect on L_2_ and ER. However, to maximise mould resistance, P/Cn8 with a higher L_2_ value was considered as the optimal preparation result in this study, and its release mechanism was investigated.

### 3.3. Release Kinetics Analysis

The release mechanism model of P/Cn can reveal the relationship between the release kinetics and the release architecture variables in essence [23]. Consequently, the release curves of P/Cn8 at 25 °C and 36 °C were plotted. Then, four kinetic models were used to fit the release curves. Figure 3 shows the fitting results of the release curves of P/Cn8 at different environment temperatures. Ultimately, the fitting effect was judged according to the fitting correlation coefficient R^2^ (Table 3), followed by the kinetic analyses.

As depicted in Table 3, at 25 °C, the fitting correlation coefficients R^2^ of the P/Cn8 release curve to the zero-order, first-order, Higuchi, and Peppas dynamics model were 0.11427, 0.96479, 0.38481, and 0.80165, respectively. At 36 °C, the corresponding fitting correlation coefficients R^2^ of the P/Cn8 release curve to the four kinetic equations were 0.24557, 0.99905, 0.55642, and 0.88923, respectively. All P/Cn8 conformed to the first-order kinetic model at 25 °C and 36 °C, and the corresponding kinetic equations were Q = 22.295 × (1 − e^−0.091t^) and Q = 88.143 × (1 − e^−0.036t^), respectively. R^2^ (36 °C) was significantly larger than R^2^ (25 °C), indicating that the higher the release temperature was, the more consistent the release mechanism was with the first-order kinetic model. On one hand, the hydrogen bond formed between PNIPAm and water may be broken at high temperatures [29], and P/Cn8 releases citral under the action of hydrophobic groups. On the other hand, the diffusion rate of citral increases with an increase in temperature.

Figure 3 (Actual) depicts that 0–128 min was the burst-release stage of citral, during which citral was rapidly released. This could be partly because citral was in the free state or at the edge of PNIPAm hydrogel, and with the increase in the temperature, the diffusion rate of free citral increased, which increased the burst-release effect [30]. The moderate release process was noted at 128–512 min, and citral was released by free diffusion under the influence of concentration gradient [31]. With the passage of time, the concentration of citral in PNIPAm decreased and the release driving force decreased [15,27]; therefore, its release rate slowed down at 512–1024 min.

In addition, the cumulative release rate of P/Cn8 at 36 °C at 1024 min was 89.986%, and the embedded citral was almost completely released. After 1024 min, the cumulative release rate at 25 °C was only 24.348%. In conclusion, PNIPAm showed excellent protective and sustained-release effects on citral.

### 3.4. Effect of Pressure

P/Cn8 was taken after dialysis when the impregnation time was 60 min and CR was 1. Bamboo strips were impregnated at different pressures; R_1_ and R_2_ were calculated, and then the two-drug loadings were plotted according to the pressure (Figure 4).

As shown in the figure, both R_1_ and R_2_ increased with an increase in the pressure. When the pressure value was 0–0.5 MPa, the amount of the two types of drug loading increased, whereas when the pressure was 0.5 MPa, R_1_ and R_2_ were 281.420 mg/m^2^ and 22.747 mg/m^2^, respectively. However, with an increase in the pressure, the amount of increase in both the drug loads decreased. R_1_ and R_2_ of the bamboo strips only increased by 2.336% and 2.171% compared with 0.5 MPa, respectively, at 0.6 MPa. Under the premise of considering the actual production benefit, 0.5 MPa, therefore, was determined as the most appropriate pressure. In addition, after 0.5-MPa pressure impregnation, R_1_ and R_2_ of bamboo strips were 318.257% and 737.518% of that under normal pressure (the pressure value of 0 MPa), respectively. This observation indicates that pressure can significantly improve the drug loading of bamboo strips. Because under the action of pressure, the air inside the bamboo was discharged, the voids inside the bamboo were increased, so that the resistance of P/Cn diffusion into the bamboo was weakened, and the permeability and adsorption performance of the bamboo were greatly improved.

### 3.5. Effect of Impregnation Time

The P/Cn8 completed by dialysis was collected, and the bamboo strips were treated at different impregnation times under the pressure of 0.5 MPa and CR of 1. R_1_ and R_2_ were calculated, and the results of the two-drug loads were plotted according to the pressure (Figure 5).

As illustrated in Figure 5, R_1_ and R_2_ both increased with the extension of impregnation time. When the impregnation time was <120 min, the two-drug loading of bamboo strips increased rapidly with time, and the curve slope increased. At impregnation time of 120 min, R_1,_ and R_2_ were 359.815 mg/m^2^ and 26.852 mg/m^2^, respectively. However, with the extension of impregnation time, the drug loading increased slowly. R_1_ and R_2_ only increased by 7.119% and 4.137% at the impregnation time of 150 min compared with that at 120 min. This result can be attributed to the fact that the surface of bamboo was saturated with P/Cn, and therefore, the increased impregnation time had only a slight effect on drug loading. Under the premise of considering the actual production efficiency, 120 min was determined as the optimum impregnation time.

### 3.6. Effect of CR

P/Cn8 after dialysis was collected and used to treat the bamboo strips under different CR, pressure of 0.5 MPa and impregnation time of 120 min. R_1_ and R_2_ were accordingly calculated, and the results of the two-drug loading were plotted according to the pressure (Figure 6).

As observed in the Figure 6, R_1_ and R_2_ exhibited different trends with the change in CR. R_1_ first showed an increasing trend and then a decreasing trend with increasing CR, while the maximum R_1_ of bamboo was 414.506 mg/m^2^ at CR = 0.6. This observation can be attributed to the fact that the greater the concentration of P/Cn, the greater the relative viscosity of its solution was, and the worse the fluidity and the obstruction of water infiltration into bamboo was. R_2_ mainly depended on the concentration of P/Cn after evaporation of water in the bamboo strips. Therefore, an increase in the concentration of P/Cn could significantly improve R_2_. However, considering that high concentration can easily cause gel precipitation that makes it difficult to impregnate bamboo [32,33], the value of 1.0 was eventually determined as the optimal CR. In other words, the original concentration of P/Cn8 was selected for bamboo strip processing. At this concentration, the R_2_ of bamboo was as high as 26.852 mg/m^2^.

In summary, the optimal process conditions for P/Cn impregnation of bamboo strips were as follows: 0.5 MPa pressure, 120 min impregnation time, and CR of 1.0. P/Cn8. The bamboo composites were prepared through this process for subsequent characterisation and mould-proof experiments.

### 3.7. FT-IR Analysis

Pure, untreated bamboo strips were used as blanks, and changes in the molecular structure of P/Cn8/bamboo were analysed through FT-IR. We found that, as per the spectrogram of blank bamboo strips shown in Figure 7, the absorption peak at wave number 1735 cm^−1^ was caused by the stretching vibration of the C=O hydroxyl groups on structures such as xylose and arabinose, which indicates the existence of hemicelluloses [34]. The absorption peaks related to the benzene ring structure at 1601, 1507, and 1244 cm^−1^ were the characteristic absorption peaks of lignin, the deformation vibration of CH_2_ in lignin at 1460 cm^−1^, and the stretching vibration of C-O in lignin at 1327 cm^−1^. The absorption peaks at 1424, 1377, and 897 cm^−1^ represented the characteristic absorption peaks of cellulose [35]. However, all peaks of P/Cn8/bamboo remained unchanged in the spectrogram, which indicated that the chemical composition of the bamboo strips did not change after P/Cn8 dipping. In addition, the characteristic absorption peak of citral did not appear in this spectrum; hence, it was speculated that citral existed in bamboo in the form of a complex and that its molecular vibration was limited by PNIPAm. Therefore, the characteristic absorption peak of the aldehyde group of citral was not shown after encapsulation [36].

### 3.8. Analysis of Mould Resistance

Based on the previous experimental results, four groups, namely PT, OT, RT, and UT, were selected to conduct indoor mould prevention experiments. During the mould control period, the infection grades of bamboo slices were recorded weekly (Table 4), while the mould control results were photographed after 28 days (as shown in Figure 8) to analyse the influence of different treatment methods on the mould control performance of the bamboo strips.

As shown in Figure 8 and Table 4, the surface of UT was covered with clearly visible PC, TV, AN, and PTA, respectively, and the infection value reached 4.0, which fully met the requirements of the mould control experiment. In addition, all the bamboo pieces were discoloured, and the bamboo itself lost colour, which reduced the bamboo’s use value. Therefore, it is necessary to treat bamboo with mould prevention. The surface of RT was also covered with clearly visible mould, and the infection grade on the surface of bamboo slices was 100%, indicating that P/Cn8 could not protect bamboo slices from mould despite sustained release.

However, the observed results of UT and RT infection with mould during the experiment showed that RT infection levels were lower than UT infection levels during the same time period (0–3 weeks). These results indicated that the P/Cn8 after sustained release of mould growth inhibition played a role, but, as a result of the reduced citral content, it could not prevent the fungus infection of bamboo. As shown in Figure 8 and Table 4, a small amount of mould remained on the surface of OT, and the corresponding infection values of PC, TV, AN, and PTA were 1.2, 2.1, 3.2, and 3.0, respectively. Correspondingly, PT was infected by all moulds at time 0 during the mould control period, and the bamboo strips remained completely natural and infused with a fresh lemon aroma. Therefore, the mould resistance of bamboo strips after P/Cn8-pressure impregnation treatment was obviously better than that of the atmospheric pressure impregnation treatment, which could be attributed to the promotion of P/Cn8 infiltration inside the bamboo tissue under pressure treatment, such that the bamboo strips could have a higher drug-loading capacity and hence a more lasting mould resistance effect.

## 4. Conclusions

In this study, a PNIPAm-supported citral sustained-release system was prepared through soap-free emulsion polymerisation, and the release mechanism, mould prevention process, and performance of the system were evaluated. The cumulative release parameters of the sustained-release system at different temperatures and time points were assessed, and the release mechanism was evaluated using a common drug-release model. The results show that the system conformed to the first-order kinetics model at 25 °C and 36 °C and that the higher the temperature, the better the first-order kinetics model match was. During the release process, PNIPAm exerted strong protective and sustained-release effects on citral. The effects of impregnating conditions on the drug-loading parameters of bamboo were investigated to reveal the following optimum conditions: pressure of 0.5 MPa, impregnation time of 120 min, and CR of 1.0. The laboratory mould control experiment results revealed that under the optimal conditions of release and impregnation time, the control efficiency of the bamboo treatment with pressure impregnation against the common bamboo moulds, such as *P. citrinum*, *T. viride*, *A. niger*, and mixed mould reached 100% after 28 days, and the original colour of the bamboo maintained during the mould control process. All in all, the relationship between the release mechanism and mildew prevention performance of the sustained-release system was fully revealed in this study, and it showed great application potential in bamboo mildew prevention. In addition, due to the good hydrophilicity of the system, and the mildew resistance of bamboo is more difficult than other substrates (bamboo has a higher density, which makes it difficult for the mildew inhibitor to penetrate), it has full potential to be applied to other hydrophilic biomass materials, such as wood and crop straw.

## Figures and Tables

**Figure 1 polymers-13-03314-f001:**
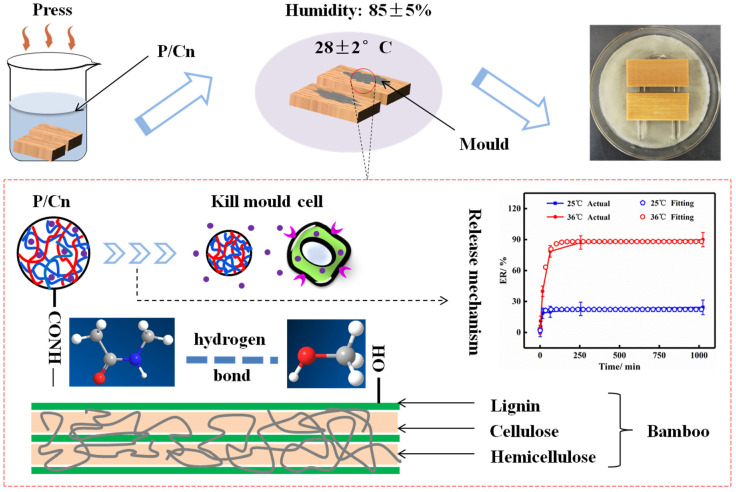
The schematic diagram of experimental principle.

**Figure 2 polymers-13-03314-f002:**
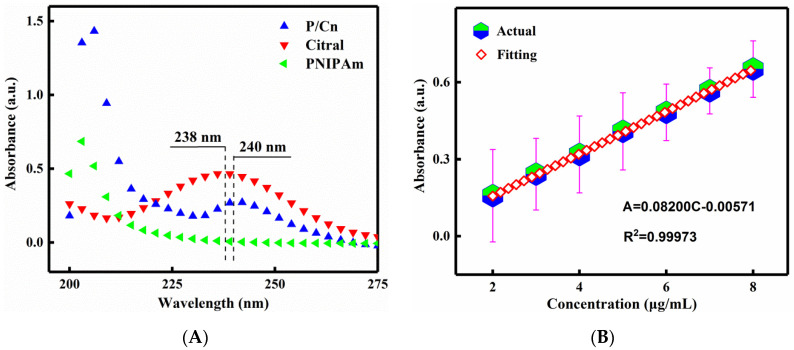
The Uv-vis spectra (**A**) and the standard curves (**B**) of P/Cn in PBS.

**Figure 3 polymers-13-03314-f003:**
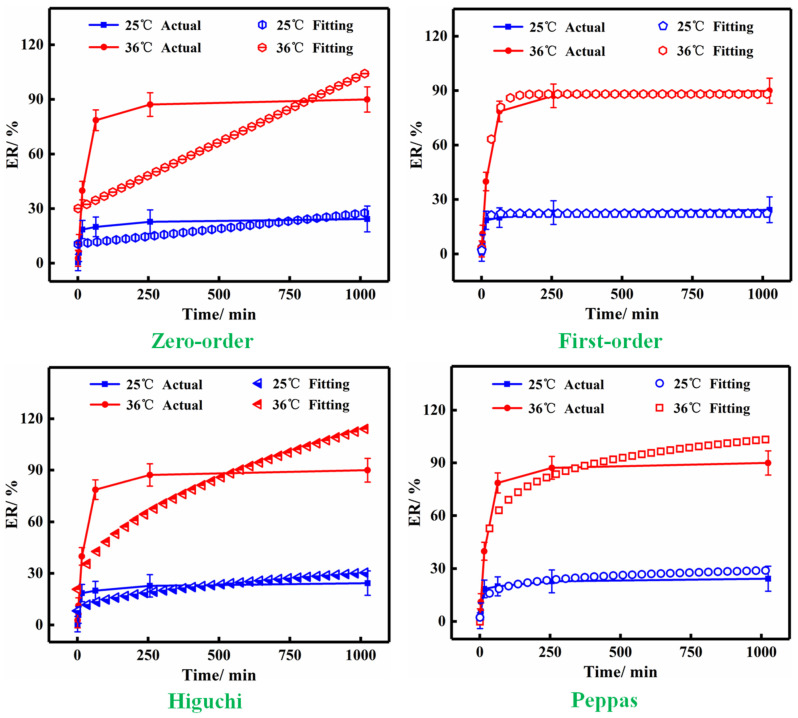
The release curves and fitting results of P/Cn8.

**Figure 4 polymers-13-03314-f004:**
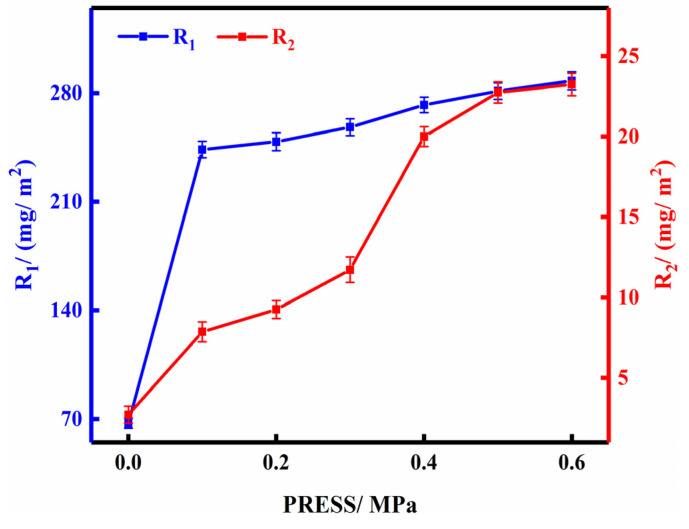
The effect of pressure on drug loading of bamboo strips.

**Figure 5 polymers-13-03314-f005:**
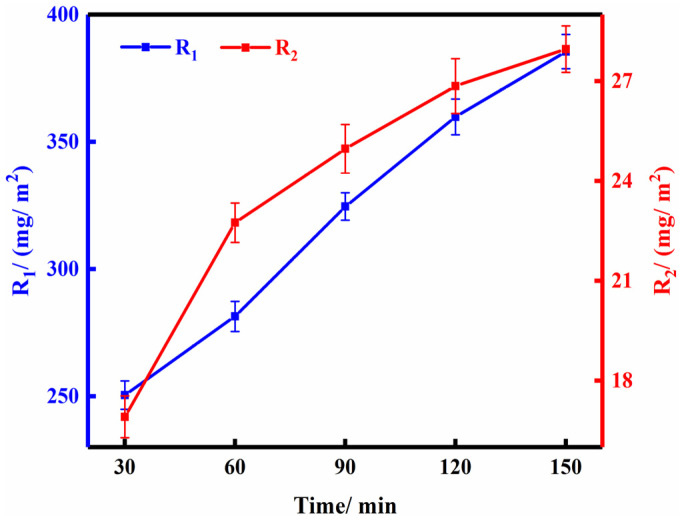
The effect of impregnation time on drug loading of bamboo strips.

**Figure 6 polymers-13-03314-f006:**
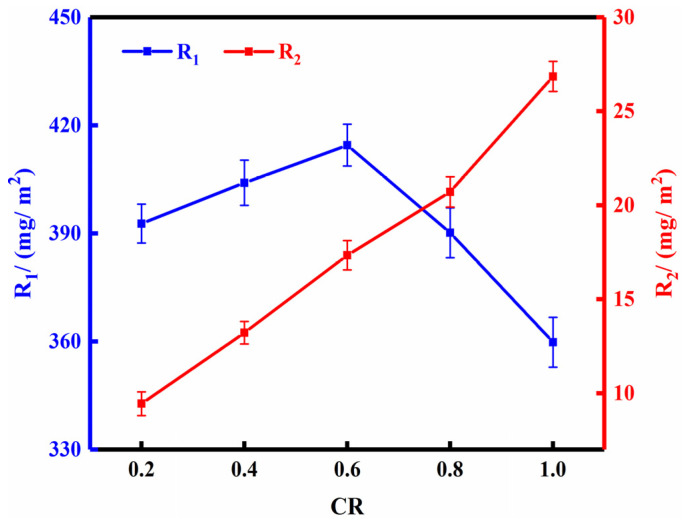
The effect of CR on drug loading of bamboo strips.

**Figure 7 polymers-13-03314-f007:**
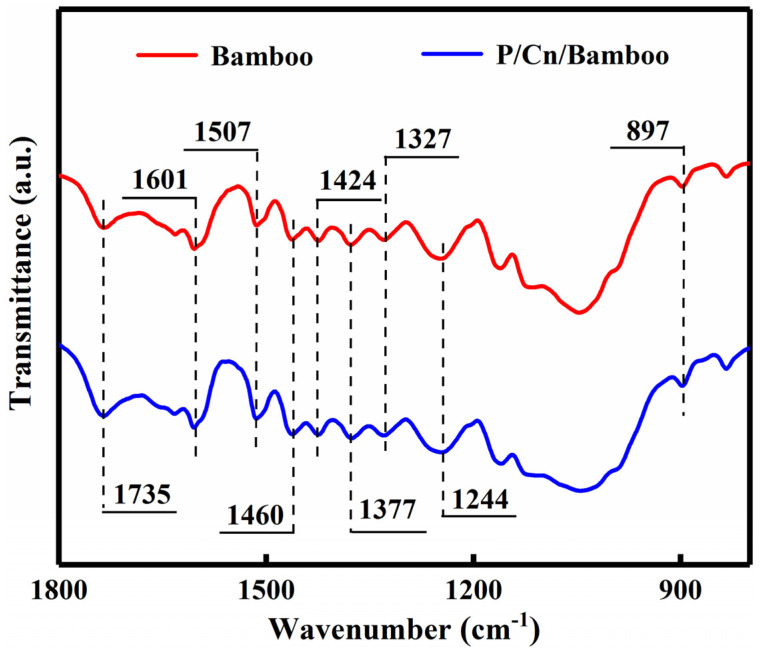
The FT−IR spectra of impregnated bamboo strips.

**Figure 8 polymers-13-03314-f008:**
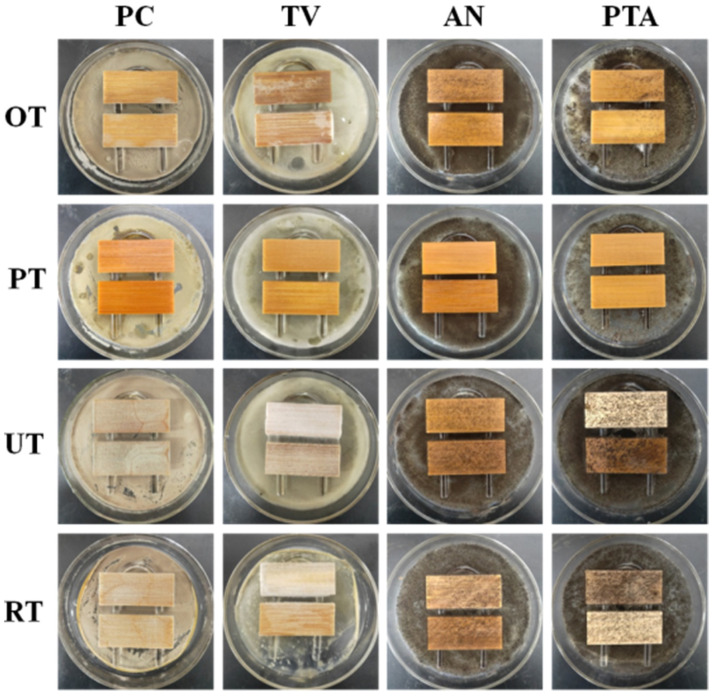
Mould control results of bamboo strips after 28 days.

**Table 1 polymers-13-03314-t001:** The results of orthogonal test.

Test No.	Factor			L_2_ (mg)	ER (%)
	A (mg)	B (h)	C (°C)		
1	1	1	1	5.099	81.919
2	1	2	2	5.628	85.675
3	1	3	3	4.942	81.919
4	2	1	2	19.820	82.893
5	2	2	3	19.748	82.197
6	2	3	1	22.515	90.542
7	3	1	3	27.132	85.953
8	3	2	1	33.469	89.986
9	3	3	2	31.093	95.828
k_1_ (L_3_)	5.223	17.35	20.361		
k_2_ (L_3_)	20.694	19.615	18.847		
k_3_ (L_3_)	30.565	19.517	17.274		
k_1_ (ER)	83.171	83.588	87.482		
k_2_ (ER)	85.211	85.953	88.132		
k_3_ (ER)	90.589	89.43	83.356		
R (L_3_)	25.342	2.265	3.087		
R (ER)	7.418	5.842	4.776		

**Table 2 polymers-13-03314-t002:** The results of ANOVA.

Source	DEVSQ		F		F_0.01_	F_0.1_	Significance	
	L_2_	ER	L_2_	ER	L_2_	ER	L_2_	ER
A	978.986	88.113	610.721	9.245	99	9	*	*
B	9.831	51.801	6.133	5.435	99	9		
C	14.296	40.253	8.918	4.223	99	9		
Error	1.600	9.530						
Total	1004.713	189.697						

*: Represent the significant influence of parameter.

**Table 3 polymers-13-03314-t003:** Kinetic fitting results.

Equation	25 °C		36 °C	
	Fitted Equation	R^2^	Fitted Equation	R^2^
Zero-order	Q = 0.017t + 10.586	0.11427	Q = 0.073t + 30.049	0.24557
First-order	Q = 22.295 × (1 − e^−0.091t^)	0.96479	Q = 88.143 × (1 − e^−0.036t^)	0.99905
Higuchi	Q = 0.719t^1/2^ + 7.528	0.38481	Q = 3.025t^1/2^+17.720	0.55642
Peppas	Q = 44,021.197t^0.00088^ − 44,018.933	0.80165	Q = 55,938.699t^0.00027^ − 55,938.810	0.88923

**Table 4 polymers-13-03314-t004:** Infection grade of bamboo strips during mould control.

Moulds	Day 7	Day 14	Day 21	Day 28	Group
PC	0.1	0.6	0.9	1.2	OT
	0	0	0	0	PT
	1.2	2.2	3.5	4.0	UT
	1.0	2.0	3.0	4.0	RT
TV	0.7	1.3	1.9	2.1	OT
	0	0	0	0	PT
	1.5	2.5	3.6	4.0	UT
	1.1	2.3	3.2	4.0	RT
AN	1.3	2.1	2.6	3.2	OT
	0	0	0	0	PT
	2.0	3.5	4.0	4.0	UT
	1.8	3.1	3.9	4.0	RT
PTA	1.4	2.2	2.6	3.0	OT
	0	0	0	0	PT
	2.1	3.6	4	4	UT
	1.9	3.6	3.9	4	RT

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
