# Peer review of "Investigation of the Release Mechanism and Mould Resistance of Citral-Loaded Bamboo Strips"

_polymers, 2021, doi:10.3390/polym13193314_

Round 1
Reviewer 1 Report
Nanohydrogel prepared using PNIPAM. PNIPAM is an interesting material, authors showed a good application using this material. The following points need to be considered to improve the manuscript:
- Please mention the temperature during preparation of PNIPAm/Citral nanohydrogels
- You used crosslinker, please mention the effect of crosslinker on property, how the crosslinker change the structure.
- 'mould and mold' both has been written in manuscript, please correct it.
- The material is useful for bamboo, is it possible to use the material for other substrates? If yes, please mention it.
- Why you choose PNIPAM for this study? mention its advantage to use it? Add this in introduction.
Reviewer 2 Report
The manuscript “Investigation of the release mechanism and mould resistance to bamboo of citral sustained-release system supported by nano hydrogel” is reviewed carefully. Peng et al. Reported a sustained-release system loading citral by using PNIPAm nanohydrogel as a carrier for mould resistance of bamboo strips. They evaluated their results with four kinetic models and found the first-order the best fit to their experiments. They resulted in optimal parameters of 0.5 MPa, immersion time of 120 min, and the system concentration ratio of 1 for achieving the best result.
The paper is well-written, except some grammatical and typos mistakes. But still the goal and application of the work are missing. The authors need to address some points and re-submit their manuscript for consideration in Polymers. My major and minor comments are below:
Major comments:
1- The objective is missing. What are you going to do with after mould resistance? Is it for food packaging application?
2- What is the novelty of the work compare to the state of the art e.g.:
- https://doi.org/10.1098/rsos.202244
- https://doi.org/10.1155/2021/5949458
- https://doi.org/10.1007/s10086-011-1223
3- The title is vague. Maybe this one is more clear and eye-catching “Investigation of the release mechanism and mould resistance of citral-loaded bamboo strips”
Minor comments:
1- There are some grammatical and typos mistakes:
- Sentences should be passive. Authors need to avoid to start their sentences with We e.g. line 11, 211, 212, and 361 etc.
- Line 13: “release models, namely” à “release models namely”, “Peppas, were” à “Peppas were”
- Line 21: “Compared with the other” à “Compared to the other”
- Line 31: “materials has been increasing” is not grammatically correct.
- Line 44: “These material have” à “These materials have”
- Line 263: “of pressurized pressure” is vague
- Line 343: “This suggesting that” has no verb. Also “slow release after” is not correct.
Line 367: “revealed that, under” à “revealed that under”
2- In line 44: what do you mean by nano-properties?
3- Figure 1 is not mentioned within the text!
4- Line 92: It is not relevant to mention “Deionised water was prepared in the laboratory.”
5- Why did you use PBS as the media?
6- You talked about “mould” within the main text, but mentioned “mold” somewhere e.g. in Figure 1. You need to keep regularity in the mnuscript.
7- Draw the standard curve is not a suitable sub-section
8- The statistical analysis (Table 2 etc.) are better to be at the end of the manuscript.
9- Line 256-261 need to be re-written.
10- The relevant samples for the line colours in Figures e.g. 4, 5, and 6 are not specified.
11- Line 281: What do you mean by CR? Abbreviations are better not to be in sections.
12- In supplementary file: The caption of Table 2 needs to provide more information.
